# GDF15 Promotes Cell Growth, Migration, and Invasion in Gastric Cancer by Inducing STAT3 Activation

**DOI:** 10.3390/ijms24032925

**Published:** 2023-02-02

**Authors:** Mina Joo, Donghyun Kim, Myung-Won Lee, Hyo Jin Lee, Jin-Man Kim

**Affiliations:** 1Department of Medical Science, College of Medicine, Chungnam National University, Daejeon 35015, Republic of Korea; 2Department of Pathology, College of Medicine, Chungnam National University, Daejeon 35015, Republic of Korea; 3Department of Internal Medicine, College of Medicine, Chungnam National University, Daejeon 35015, Republic of Korea; 4Infection Control Convergence Research Center, College of Medicine, Chungnam National University, Daejeon 35015, Republic of Korea

**Keywords:** gastric cancer, GDF15, STAT3, progression

## Abstract

Growth differentiation factor 15 (GDF15) has been reported to play an important role in cancer and is secreted and involved in the progression of various cancers, including ovarian cancer, prostate cancer, and thyroid cancer. Nevertheless, the functional mechanism of GDF15 in gastric cancer is still unclear. Immunohistochemical staining was performed to estimate the expression of GDF15 in 178 gastric cancer tissues. The biological role and action mechanism of GDF15 were investigated by examining the effect of GDF15 knockdown in AGS and SNU216 gastric cancer cells. Here, we report that the high expression of GDF15 was associated with invasion depth (*p* = 0.002), nodal involvement (*p* = 0.003), stage III/IV (*p* = 0.01), lymphatic invasion (*p* = 0.05), and tumor size (*p* = 0.049), which are related to poor survival in gastric cancer patients. GDF15 knockdown induced G0/G1 cell cycle arrest and remarkably inhibited cell proliferation and reduced cell motility, migration, and invasion compared to the control. GDF15 knockdown inhibited the epithelial–mesenchymal transition by regulating the STAT3 phosphorylation signaling pathways. Taken together, our results indicate that GDF15 expression is associated with aggressive gastric cancer by promoting STAT3 phosphorylation, suggesting that the GDF15-STAT3 signaling axis is a potential therapeutic target against gastric cancer progression.

## 1. Introduction

Gastric cancer is the fourth most common type of cancer worldwide and the second leading cause of cancer-related death [1]. Most gastric cancers are adenocarcinoma with a variety of molecular and histological subtypes affected by many environmental and genetic factors [2,3]. If gastric cancer is not diagnosed early, therapy and overall prognosis are still unsatisfactory, and the 5 years survival rate is less than 40% [4,5,6]. Moreover, tumor formation and development of gastric cancer are multifactorial tumorigenic processes and have been shown to contribute to genetic and epigenetic changes in gastric cancer progression [7,8,9]. Nevertheless, molecular mechanisms concerned with the incidence and development of gastric cancer, remain unclear. Thus, it is necessary to identify biomarkers that can predict prognosis and guide treatment through more profound insight into the molecular mechanisms of gastric cancer. 

Growth differentiation factor 15 (GDF15) is a member of the transforming growth factor β (TGF-β) superfamily, also known as macrophage inhibitory cytokine 1 (MIC-1) [10]. GDF15 is expressed in the cytoplasm as a full-length form 35-kDa precursor protein that is cleaved to produce a mature 17-kDa secreted cytokine [11,12,13]. Under normal physiological conditions, GDF15 expression is stable and weak in most tissues. However, GDF15 is remarkably upregulated under pathological conditions such as inflammation, injury, and carcinoma. Furthermore, increased circulatory levels of GDF15 are clinically related to cancer progression and chemotherapy resistance in ovarian, prostate, breast, and colorectal cancers [14,15,16,17]. This suggests that GDF15 can act as a biomarker or predictor of therapeutic resistance to gastric cancer [11,18]. Nevertheless, the role of GDF15 signaling in gastric cancer is unclear. Consequently, it is important to study the role of GDF15 to identify potential therapeutic targets that can inhibit tumor progression in gastric cancer cells.

Signal transducer and activator of transcription (STAT) proteins are transcription factors associated with solid tumor development and progression [19,20]. In particular, the signal transducer and activator of transcription 3 (STAT3) is a critical transcription factor and plays an important role in gene regulation [21,22,23]. Phosphorylated STAT3 is dimerized and translocated to the nucleus, promoting the transcription of a number of target genes, most of which are oncogenes [24]. STAT3 phosphorylation is usually abnormally activated in malignant tumors and is associated with poor clinical prognosis and many biological processes, especially cell differentiation, proliferation, and invasion [25,26,27,28]. Thus, STAT3 is an attractive target for the development of new therapeutic targets in cancer [29,30]. Given the findings that GDF15 is required for the activation of STAT3 and tumorigenesis in thyroid cancer and glioma stem cells [31,32], we focused whether GDF15 function is mediated through STAT3 signaling in gastric cancer cells. In this study, we investigated the biological function of GDF15 and its effect in gastric cancer progression.

## 2. Results

### 2.1. GDF15 Expression in Human Gastric Normal Tissue and Gastric Cancer Tissue and Serum

We investigated that the expression of GDF15 in the GEPIA data set was significantly upregulated in the gastric cancer group compared to the normal group (Figure 1A). Moreover, we examined ten pairs of gastric cancer tissues and adjacent non-malignant gastric tissues for GDF15 protein levels using Western blotting analysis (Figure 1B). The expression of GDF15 mRNA was also validated in normal gastric tissues and gastric cancer tissues by RT-qPCR (Figure 1C and Appendix A). We further analyzed serum GDF15 levels in normal gastric and gastric cancer patients (Figure 1D). We investigated the function of GDF15 in gastric cancer progression and evaluated GDF15 expression in tumor specimens from 178 patients with gastric cancer. Immunohistochemical analysis revealed that GDF15 was located in the membrane and/or cytoplasm, with varying degrees of staining intensity: negative staining (score 0), 22 cases; weak staining (score 1), 49 cases; moderate staining (score 2), 61 cases; and strong staining (score 3), 46 cases (Figure 1E and Appendix A). Moreover, clinicopathological characteristic analysis indicated that higher GDF15 expression was positively associated with invasion depth (*p* = 0.002), nodal involvement (*p* = 0.003), stage (*p* = 0.010), lymphatic invasion (*p* = 0.050), and tumor size (*p* = 0.049) (Table 1). Univariate and multivariate analyses were performed to investigate the clinical consequence of various prognostic factors that may affect patient survival. In the univariate analysis, age (*p* = 0.001), invasion depth (*p* < 0.001), nodal involvement (*p* = 0.005), stage (*p* < 0.001), tumor size (*p* < 0.001), and GDF15 expression (*p* = 0.006) were significant risk factors affecting survival in gastric cancer patients. Furthermore, multivariate analysis showed that age (*p* = 0.001), stage (*p* = 0.001), and GDF15 expression (*p* = 0.048) were prognostic factors for survival in gastric cancer patients (Table 2). GDF15 was an independent predictor of reduced overall survival in patients with gastric cancer. Kaplan–Meier analysis showed a significant association between GDF15 levels and poor survival outcomes in gastric cancer patients in the CNUH (Figure 1F) and GEPIA datasets (Figure 1G). Therefore, these results suggest that GDF15 expression is related to gastric cancer progression.

### 2.2. Knockdown of GDF15 Inhibits Cell Growth by Inducing Cell Cycle Arrest

To investigate how GDF15 regulates tumor progression in gastric cancer, stable cell lines (AGS and SNU216) were established by GDF15 knockdown, and knockdown of GDF15 was confirmed by RT-PCR and Western blotting analysis (Figure 2A). GDF15-knockdown cells showed significantly reduced cell proliferation and colony formation compared to shCtrl cells (Figure 2B,C). The percentage of cells in the G0/G1 phase of the cell cycle remarkably increased after knockdown, whereas the percentage of cells in the S and G2/M phases decreased in both GDF15-knockdown cell lines (Figure 2D and Appendix A). Western blotting analysis also showed that the cell cycle regulatory proteins such as CDK2, CDK4, CDK6, cyclin D, and cyclin E were downregulated following GDF15 knockdown (Figure 2E). These findings indicate that GDF15 knockdown inhibits cell growth by inducing cell cycle arrest in gastric cancer cells.

### 2.3. Knockdown of GDF15 Suppresses Cell Migration and Invasion

To confirm the effect of GDF15 on cell motility and invasion, we evaluated using gap closure and transwell assays. GDF15-knockdown cells showed significantly reduced motility in gap closure assays compared to shCtrl cells (Figure 3A). GDF15-knockdown cells showed remarkably decreased cell migration and invasion compared to shCtrl cells (Figure 3B,C). In addition, the expression of epithelial markers such as E-cadherin and Ep-cam was upregulated in GDF15-knockdown cells, whereas the expression of mesenchymal markers such as vimentin, Slug, and Twist was downregulated compared to shCtrl cells (Figure 3D). Phalloidin staining showed that GDF15-knockdown cells significantly reduced stress fiber formation compared to shCtrl cells (Figure 3E). These results demonstrate that knockdown of GDF15 inhibits gastric cancer cell motility, migration, and invasion.

### 2.4. GDF15 Knockdown Suppresses STAT3 Activation

We next investigated the mechanisms by which GDF15 induces carcinogenesis in gastric cancer cells. Previous studies reported that GDF15 is required for the activation of STAT3 and tumorigenesis in thyroid cancer and glioma stem cells [31,32]. We thus examined whether the blockade of GDF15 affected the STAT3 activation in gastric cancer cells. We observed that GDF15 knockdown decreased the phosphorylation of STAT3 and c-MYC (Figure 4A). In addition, there are several signaling pathways including AKT, ERK, p38, and NF-κB that are regulated by GDF15 in other cancers [33,34]. When we further screened whether these signaling pathways are perturbed in GDF15-knockdowned gastric cancer cells, we found that the activation of AKT, ERK, p38, and NF-κB was not different between shCtrl and GDF15-knockdown gastric cancer cells (Appendix A). These data suggest that GDF15-mediated tumorigenesis is mainly mediated through STAT3, but not these signaling pathways in gastric cancer cells. Furthermore, to determine the association between GDF15 and STAT3 in gastric cancer, we analyzed their correlation in the GEPIA dataset (Appendix A). The data suggested that GDF15 expression is associated with STAT3 expression in gastric cancer. These findings suggest that the GDF15-STAT3 signaling axis regulates tumor progression. To determine whether GDF15 regulates STAT3 expression, we added IL-6, a growth factor and inducer of STAT3 phosphorylation, to GDF15-knockdown cells (Figure 4B). IL-6 treatment remarkably increased the phosphorylation of STAT3. Additionally, cell cycle regulatory proteins such as CDK2, CDK4, CDK6, cyclin D, and cyclin E levels were increased by IL-6 in GDF15-knockdown cells (Figure 4C). Furthermore, IL-6 increased cell migration and invasion and reversed the effect of GDF15 knockdown (Figure 4D,E). To determine whether the expression of STAT3 and its downstream factors is regulated by GDF15, we transiently transfected siSTAT3 to GDF15-overexpressing cell lines (Appendix A). Furthermore, to determine whether STAT3 activity was required for GDF15-mediated migration and invasion, we transiently transfected siSTAT3 into GDF15-overexpressing cell lines (Appendix A). These results demonstrate that GDF15 regulates cell migration and invasion through STAT3 in gastric cancer cells.

### 2.5. GDF15 Overexpression Promotes Cell Proliferation, Migration, and Invasion by Upregulating STAT3 Signaling

To evaluate the role of GDF15 in the proliferation, migration, and invasion of gastric cancer cells, we established GDF15-overexpressing cell lines (Figure 5A and Appendix A). GDF15 overexpression increased cell proliferation and colony formation in gastric cancer cells compared to that in mock cells (Figure 5B and Appendix A). STAT3 phosphorylation and c-MYC levels were remarkably increased in GDF15-overexpressing cells (Figure 5C). In addition, GDF15 overexpression also increased the expression of the cell cycle regulatory proteins such as CDK2, CDK4, CDK6, cyclin D, and cyclin E (Figure 5D). GDF15 overexpression also remarkably increased cell migration and invasion (Figure 5E,F). In addition, GDF15-overexpressing cells exhibited increased motility in the gap closure assay (Appendix A). GDF15-overexpressing cells almost closed wide gaps compared to mock cells. These results reveal that GDF15 promotes cellular motility and invasion via active STAT3 signaling.

## 3. Discussion

Gastric cancer is one of the most common cancers with a high degree of malignancy; however, despite the development of diagnostic tools and antitumor drugs, the survival time of gastric cancer patients has not been increased significantly [35]. In addition, the lack of effective biomarkers for the early detection of gastric cancer means that patients are usually diagnosed at an advanced stage [36]. Therefore, it is necessary to discover new biomarkers for the early diagnosis of gastric cancer. Recently, GDF15 has been reported as a stress-response cytokine. GDF15 expression levels can increase suddenly in response to various cellular stress signals, such as hypoxia, inflammation, and different malignancies. Additionally, elevated levels of circulating GDF15 have been discovered in several malignant tumors and are considered potential tumor biomarkers [14,34]. In this study, we observed that GDF15 expression significantly promoted gastric cell growth, migration, and invasion. 

GDF15 is a widely overexpressed protein in human cancers and has been extensively investigated as a regulator of many biological processes [34,37,38]. Furthermore, GDF15 secretion generally increases in response to cellular stress or damage, and secreted GDF15 proteins are found in the extracellular matrix and human blood [38,39]. Thus, the GDF15 level has recently been suggested as a predictive biomarker for recurrence and survival in colorectal cancer [40] and prostate cancer [41]. From clinical data, we observed that GDF15 levels were positively correlated with invasion depth, nodal involvement, stage, lymphatic invasion, and tumor size. Furthermore, serum GDF15 levels were remarkably higher in the gastric cancer group than in the normal gastric group. We discovered that the high expression of GDF15 was related to gastric cancer progression and elevated serum GDF15 levels in gastric cancer patients. From our results, we speculated that GDF15 might be related to the migration and invasion of gastric cancer cells. Additionally, GDF15 knockdown in gastric cancer cells considerably reduced cell proliferation by inducing G0/G1 cell cycle arrest. Our data suggested that GDF15 was associated with the expression of G0/G1 phase-regulating proteins such as CDK2, CDK4, CDK6, cyclin D, and cyclin E. The knockdown of GDF15 also decreased cell motility, migration, and invasion, and these results were reversed through GDF15 overexpression. Consequently, it is necessary to understand the molecular mechanism of GDF15 in gastric cancer.

GDF15 stimulates the PI3K/AKT, ERK, and NF-κB signaling pathways in various cancers [39,42,43]. However, the mechanism by which GDF15 contributes to the tumor progression of gastric cancer is not well defined. Our study revealed that at the molecular level, GDF15 regulated cell proliferation, migration, and invasion by upregulating STAT3 phosphorylation. According to our data, GDF15 knockdown inhibited STAT3 activation, but not AKT, ERK, p38, and NF-κB activation. Many studies have found that the knockdown of STAT3 induces G0/G1 cell cycle arrest and apoptosis and affects cell growth. Cell cycle regulation is very important for cell progression and survival; however, abnormal cell cycle progression in malignant cells promotes tumorigenesis and prevents cell death [44,45]. In addition, cell cycle progression rigorously monitors healthy cell proliferation and progression checkpoints during phase transitions [46,47]. The loss of cell cycle control is a typical hallmark of tumorigenesis. In our results, we found that GDF15 inhibition induced cell cycle arrest in gastric cancer cells by inhibiting STAT3 activation. IL-6 treatment also increased STAT3 phosphorylation and cell-cycle-regulating proteins, such as MYC in GDF15-knockdown cells. IL-6, an inflammatory tumor cytokine, activates a series of downstream factors by activating the IL-6/STAT3 signaling pathway, which plays a significant role in the growth and development of many human cancers [19,48]. MYC drives the multiple synthetic functions required for rapid cell division. Additionally, a multifunctional transcription factor inhibits the expression of genes with antiproliferative functions in human cancers [49,50]. MYC, bound to the origins of replication, induces DNA replication by upregulating genes encoding proteins required for replication initiation [51].

Epithelial–mesenchymal transition (EMT) is a key reversible process by which cancer cells transition to a highly migratory and invasive phenotype regulated by epigenetic mechanisms such as DNA methylation, acetylation, histone methylation, and microRNA binding [52,53,54]. Furthermore, STAT3 activation is also related to EMT. The present study demonstrated that GDF15-mediated STAT3 activation induced EMT in gastric cancer. The knockdown of GDF15 reduced cell migration, invasion, and EMT marker expression; however, these effects were reversed by GDF15 overexpression. Although our data has a limitation of the lack of in vivo models, clinical data and in vitro studies strongly suggest that GDF15 promotes the progression of gastric cancer. In addition, our study demonstrated that GDF15 is involved in cell proliferation, migration, and invasion in gastric cancer via STAT3/MYC signaling. Future studies are needed to characterize in vivo function of GDF15 in tumorigenesis of gastric cancer.

## 4. Materials and Methods

### 4.1. Clinical Specimens

A total of 178 gastric cancer tissues were obtained from Chungnam National University Hospital (CNUH) (Daejeon, Republic of Korea). All samples used in this study were approved by the CNUH ethics committee (CNUH-IRB 2021-03-090). Clinicopathological characteristics including sex, age, stage, lymph node metastasis, and lymphovascular tumor invasion were evaluated. The patients’ stage was determined according to the tumor node metastasis (TNM) staging system and all samples were subjected to pathological analysis; this study was performed according to the Declaration of Helsinki and the Good Clinical Practice guidelines.

### 4.2. Tissue Microarray Construction

A tissue microarray was constructed using formalin-fixed, paraffin-embedded (FFPE) tissue blocks obtained from 178 gastric cancer patients. For each tumor, the typical tumor regions were carefully selected from hematoxylin and eosin (H&E)-stained sections of a donor paraffin block. Each case was marked with two cylindrical cores (2 mm in diameter) from a tumor, which were punched using an automated tissue microarrayer (3DHistech, Budapest, Hungary). Tissue microarrays were generated using the TMA Grand Master system (3DHistech) following the manufacturer’s instructions.

### 4.3. Specimen Preparation and Immunohistochemical Analysis

Specimen preparation and immunohistochemical analysis were performed according to our previously reported protocol (26). For Western blot analysis and mRNA expression analysis, we extracted cancer tissue samples from gastric cancer patients and paired adjacent non-cancer tissue samples. The Western blot analysis samples are extracted using the PRO-PREP™ Protein Extraction Solution (17081; iNtRoN, Seongnam, Republic of Korea) and total RNA was extracted using the easy-spin Total RNA Extraction Kit (17221; iNtRoN). All samples were collected according to the manufacturer’s protocol, and experiments were performed in clean conditions, and the equipment was pre-autoclaved. For immunohistochemical analysis, 3 μm-thick sections were cut from the recipient blocks. Briefly, the sections were dewaxed in xylene and rehydrated in graded alcohols. Sections were washed with water prior to antigen retrieval using a Dako PTLink machine (Dako, Glostrup, Denmark) with 10 mM of sodium citrate buffer (pH 6.0). The sections were treated with 3% hydrogen peroxide to block endogenous peroxidase and pre-incubated with a serum-free protein block solution (Dako) to remove background staining. The polyclonal mouse antibody raised against human GDF15 (NBP1-81050; Novus Biologicals, Littleton, MA, USA) was diluted 1:200 in background-reducing diluents (Dako). The slides were then incubated with EnVision anti-mouse (Dako) polymer. The reaction products were visualized using diaminobenzidine plus substrate–chromogen solution. The slides were counterstained with Meyer’s hematoxylin and then mounted. Careful rinsing with multiple changes of phosphate buffered saline (PBS) was performed between procedure steps. Samples without primary antibody or samples containing pre-immune IgG1 were used as negative controls to estimate nonspecific staining.

### 4.4. Evaluation of Immunohistochemical Staining

Immunohistochemical staining was performed as described previously (26). Staining results were estimated by two independent pathologists (JMK and DK) blinded to the clinicopathological details of the patients. According to staining intensity, immuno-histochemical findings were scored as follows: 0, no staining; 1, weak staining; 2, moderate staining; and 3, strong staining. For heterogeneous staining within a sample, a higher score was selected if more than 50% of cells exhibited a higher staining intensity. The mean score across all patients was obtained by averaging the scores of both tumor cores from the same patient. Cases in which the staining intensity score was 0–1 were placed in the GDF15 low-expression group, and cases in which the staining intensity score was 2–3 were placed in the GDF15 high expression group.

### 4.5. Cell Culture

The human gastric cancer cell lines AGS (KCBL No. 21739) and SNU216 (KCBL No. 00216) were purchased from the Korean Cell Line Bank (Seoul, Republic of Korea). AGS and SNU216 cells were grown in RPMI1640 medium (Corning, NY, USA) supplemented with 10% fetal bovine serum and 1× penicillin/streptomycin. Cells were incubated at 37 °C in humidified air with 5% CO_2_. 

### 4.6. Reagents and Antibodies

Antibodies against GDF15 (HPA011191; Sigma, St. Louis, MO, USA), GAPDH (sc-25778; Santa Cruz Biotechnology, Santa Cruz, CA, USA), CDK2 (2546; Cell Signaling Technology, Danvers, MA, USA), CDK4 (12790; Cell Signaling Technology), CDK6 (13331; Cell Signaling Technology), cyclin D (2978; Cell Signaling Technology), cyclin E (sc-377100; Santa Cruz Biotechnology), E-cadherin (3195; Cell Signaling Technology), Ep-CAM (sc-25308; Santa Cruz Biotechnology), vimentin (3932; Cell Signaling Technology), Slug (9585; Cell Signaling Technology), Twist (sc-81417; Santa Cruz Biotechnology), phosphorylated-STAT3 (p-STAT3; 9134; Cell Signaling Technology), STAT3 (9132; Cell Signaling Technology), and c-MYC (sc-40; Santa Cruz Biotechnology) were used in Western blotting and immunofluorescence analyses. Interleukin 6 (IL-6; 200-06; Peprotech, Rocky Hill, NJ, USA) was resuspended in the medium and used to treat the cells.

### 4.7. Knockdown of GDF15 in Gastric Cancer Cell Lines

GDF15 knockdown in gastric cancer cells was achieved through the lentivirus-mediated transduction of GDF15 siRNA with the pLKO.1-puro lentiviral vector (Clontech, Mountain View, CA, USA). For stable transfection in HEK-293T (Clontech) cells, lentiviral vectors were co-transfected with the virus mix (Sigma) using Lipofectamine 3000 (L3000015; Invitrogen, Carlsbad, CA, USA). The supernatant medium containing the virus was harvested and concentrated using a Lenti-X-Concentrator (Clontech) and virus was added to AGS and SNU216 cells with 5 µg/mL polybrene (Santa Cruz Biotechnology). Thereafter, the medium was replaced with fresh medium containing puromycin (Sigma)and cultured in the presence of puromycin for 2 weeks to identify puromycin-resistant clones.

### 4.8. Overexpression of GDF15 in Gastric Cancer Cell Lines

For the stable overexpression of GDF15, a plasmid-encoding full-length GDF15 was cloned from pLenti-C-mGFP-P2A-Puro-GDF15 (Origene, Rockville, MD, USA) into the pLVX-puro lentiviral vector (Clontech). For stable transfection in HEK-293T (Clontech) cells, lentiviral vectors were co-transfected with the virus mix (Sigma) using Lipofectamine 3000 (Invitrogen). The supernatant medium containing the virus was harvested and concentrated using a Lenti-X-Concentrator (Clontech) and the virus was added to AGS and SNU216 cells with 5 µg/mL polybrene (Santa Cruz Biotechnology). Thereafter, the medium was replaced with fresh medium containing puromycin (Sigma), and cultured in the presence of puromycin for 2 weeks to identify puromycin-resistant clones.

### 4.9. Reverse Transcription PCR (RT-PCR) Analysis

RT-PCR analysis was performed according to a previously reported protocol (27). Briefly, total RNA was isolated using TRIzol reagent (Thermo Fisher Scientific, Waltham, MA, USA) according to the manufacturer’s instructions. First-strand cDNA was prepared from an RNA template using the cDNA qPCR RT Master Mix (Toyobo, Osaka, Japan), and RT-RCR was performed using the EmeraldAmp Master Mix (TaKaRa Bio, Kusatsu, Japan). The primers and oligonucleotide sequences are listed in Appendix A. Gels were visualized and analyzed using the GelDoc Xr system (Bio-Rad, Hercules, CA, USA), and band sizes and molecular weights were determined using Image Lab analysis software version 4.1 in relation to a 100 bp DNA ladder (Bioneer, Daejeon, Republic of Korea). Quantitative PCR (qPCR) was performed on the Rotor-Gene Q 2plex system (9001620; Qiagen, Hilden, Germany) using primers for the indicated genes and SYBR Green Master Mix (218073; Qiagen) following the manufacturer’s instructions.

### 4.10. Western Blotting Analysis

Western blotting was performed according to a previously reported protocol (27). Briefly, the cells were lysed in RIPA buffer (LPS Solution, Daejeon, Republic of Korea) with a protease inhibitor cocktail (Sigma) and phosphatase inhibitor cocktail (Roche, Basel, Switzerland). Cell lysates were resolved by SDS-PAGE and transferred to polyvinylidene fluoride membranes (Pall Corp., Port Washington, NY, USA). The membranes were blocked with ProNA™ General-block solution (Translab, Daejeon, Republic of Korea) and incubated overnight with primary antibodies at 4 °C. After washing, the membranes were incubated with horseradish peroxidase-conjugated anti-rabbit IgG (1:5000, cat. no. 7074; Cell Signaling Technology) and anti-mouse IgG (1:5000, cat. no. 7076; Cell Signaling Technology) antibodies diluted in the blocking solution. Immunoreactive polypeptides were detected using an improved chemiluminescence substrate (Advansta, Menlo Park, CA, USA).

### 4.11. In Vitro Cell Proliferation Assay

The cell proliferation assay was performed according to our previously reported protocol (27). Briefly, cell proliferation was measured using a Cell Counting Kit-8 (CCK-8; Dojindo Molecular Technologies, Rockville, MD, USA). Cell proliferation was measured every 24 h for 4 days, and absorbance was measured at 450 nm (Molecular Devices, San Jose, CA, USA). For the clonogenic assay, after formation of the appropriate colony size, the plates were rinsed with PBS and fixed with 10% formalin. The colonies were then stained with 0.1% crystal violet. To determine the relative number of colonies, crystal violet was eluted using 70% alcohol and the absorbance was measured at 595 nm using a spectrophotometer (Molecular Devices).

### 4.12. Gap Closure Assay

The gap closure assay was performed according to our previously reported protocol (27). Briefly, cells were seeded on each side of the chamber (Ibidi, Munich, Germany) using culture inserts for live cell analysis. After the cells were cultured, the culture inserts were removed, and the cells were further incubated with culture medium. 

### 4.13. Migration and Invasion Assays

Migration and invasion assays were performed according to a previously reported protocol (27). Briefly, migration and invasion assays were performed using 8 µm pore Transwell chambers (Corning). For the migration assay, the lower chamber was coated with 0.1% gelatin (Sigma), and for the invasion assay, the upper chamber was coated with 25 µg/mL Matrigel (BD Biosciences, San Jose, CA, USA). Serum-free medium and the cells were seeded in each well of the upper chamber, and a culture medium containing 10% fetal bovine serum was added to the lower chamber. After incubation for 24 h, the chamber on the upper surface of the inner membrane was removed using a cotton swab, and the cells were then fixed with 10% formalin and stained with 0.1% crystal violet. The number of cells that passed through the chamber was counted under a microscope (IX71; Olympus, Tokyo, Japan). Values representing the degree of migration and invasion were averaged in three independent experiments.

### 4.14. Immunofluorescence Assay

The immunofluorescence assay was performed according to a previously reported protocol (27). Briefly, cells were seeded in a chamber slide (Lab-Tek II; Thermo Fisher Scientific) and incubated, fixed with 10% formalin, permeabilized with 0.5% Triton X-100, and then blocked in blocking buffer (4% BSA in PBS). Cells were stained with anti-GDF15 antibody, washed with PBS, and treated with a secondary antibody. Finally, nuclei were counterstained with DAPI (Vector Laboratories, Burlingame, CA, USA), and coverslips were mounted on the slides. To analyze the actin cytoskeleton, cells were stained with rhodamine-conjugated phalloidin (Sigma) and washed three times with PBS. The nuclei were stained with DAPI (Vector Laboratories).

### 4.15. Cell Cycle Analysis

Cell cycle analysis was performed according to a previously reported protocol (27). Briefly, cells were seeded in a 6-well plate and incubated and fixed in 70% ice-cold ethanol. The cells were incubated with FxCycle™ PI/RNase Staining Solution (Invitrogen) according to the manufacturer’s instructions. Fluorescence was analyzed with FACS Canto II (BD Biosciences) using DiVa 6.1.1.

### 4.16. ELISA

The human GDF15 ELISA kit (DGD150; R&D Systems, Minneapolis, MN, USA) was used to determine the concentration of GDF15 in serum samples. ELISA was performed according to the manufacturer’s instructions.

### 4.17. GEPIA Database

Gene Expression Profiling Interactive Analysis (GEPIA) database analysis was performed using our previously reported protocol (27). Briefly, the GEPIA database (http://gepia.cancer-pku.cn/, accessed on 28 January 2022), a newly developed web-based tool, was used for analysis based on the Cancer Genome Atlas (TCGA) and genotype–tissue expression data. *p*-values are indicated by asterisks in GEPIA results.

### 4.18. Statistical Analysis

Each experiment was performed in triplicate to estimate the mean ± standard deviation (SD). All statistical analyses were performed using Prism (GraphPad Software, v. 5.01; GraphPad Software, Inc., San Diego, CA, USA). To determine the significance of difference between two groups, an unpaired two-tailed Student’s *t*-test was applied. One-way ANOVA followed by Dunnett’s test was used to compare multiple data. Minimum statistical significance was set at *p* < 0.05 in all cases.

## 5. Conclusions

In summary, our findings show that GDF15 functions in tumorigenesis in gastric cancer patients. Moreover, GDF15 promotes cancer progression via the phosphorylation of STAT3 and cell-cycle-related proteins in gastric cancer. GDF15 knockdown significantly decreases migration and invasiveness in gastric cancer cells. Consequently, our study suggests that the GDF15/STAT3 signaling axis may be an effective prognostic target for patients with gastric cancer.

## Figures and Tables

**Figure 1 ijms-24-02925-f001:**
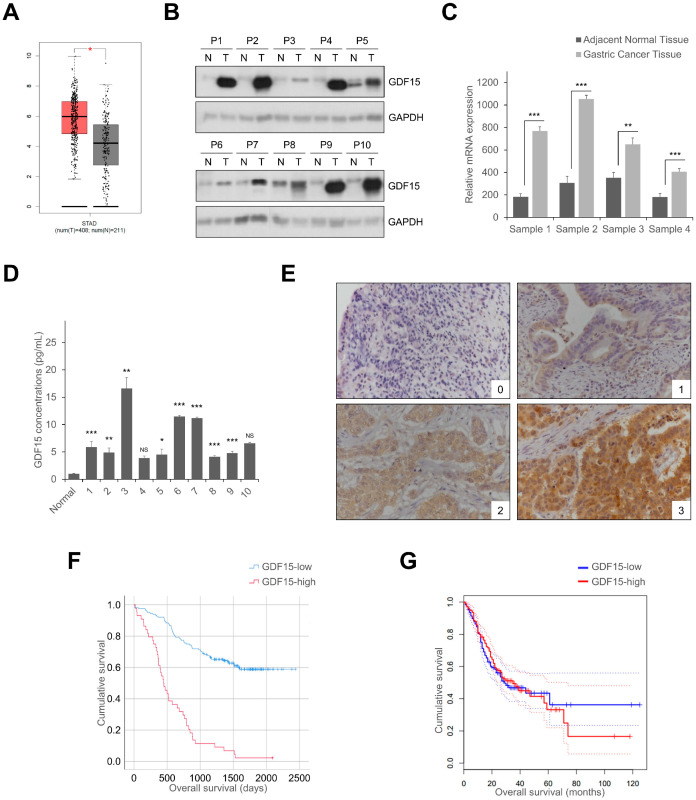
GDF15 expression in human gastric normal tissue and gastric cancer tissue and serum. (**A**) GDF15 is highly expressed in gastric cancer tumors compared to gastric normal in the GEPIA database. (**B**) Western blotting analysis of GDF15 expression levels in adjacent normal tissues and gastric cancer tissues from CNUH patients. (**C**) The mRNA expression level of GDF15 in adjacent normal tissue and gastric cancer tissues was determined by RT-qPCR. The data are presented as the means ± SD and were evaluated using Student’s *t*-test (*n* = 3). (**D**) Gastric cancer serum and normal serum were subjected to ELISA for measurement of the GDF15 level. The data are presented as the means ± SD and were evaluated using Student’s *t*-test (*n* = 3). (**E**) Expression of GDF15 protein in gastric cancer tissues. Gastric cancer tissues were immunohistochemically stained with an anti-GDF15 antibody. Magnification: ×100. 0, no staining; 1, weak staining; 2, intermediate staining; 3, strong staining. (**F**,**G**) Kaplan–Meier survival analysis revealed that overall survival is poorer in gastric cancer patients with a high expression of GDF15 among the CNUH and GEPIA datasets. NS, not significant. * *p* < 0.05, ** *p* < 0.01, and *** *p* < 0.001.

**Figure 2 ijms-24-02925-f002:**
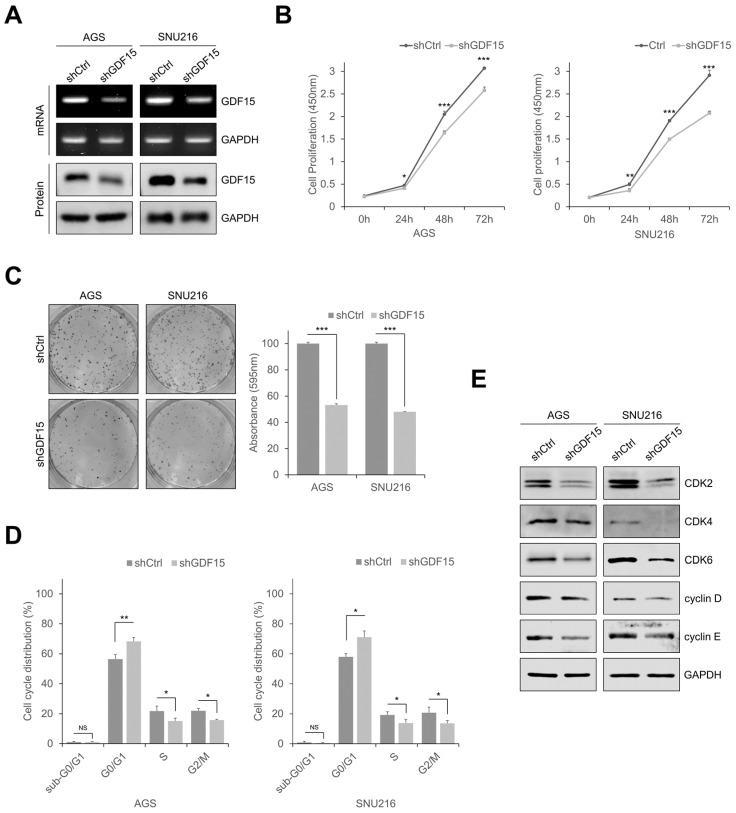
Knockdown of GDF15 inhibits cell growth by inducing cell cycle arrest. (**A**) Western blotting and RT-PCR analyses of GDF15 in GDF15-knockdown and shCtrl cells. (**B**) Cell proliferation rate as measured by cell counting analysis in GDF15-knockdown and shCtrl cells, and absorbance was measured at 450 nm using a spectrophotometer. Data are presented as the means ± SD and were evaluated using Student’s *t*-test (*n* = 3). (**C**) The colonogenic assay was performed on a 6-well culture plate in GDF15-knockdown and shCtrl cells. Cells stained with crystal violet were dissolved in 70% alcohol, and absorbance was measured at 595 nm using a spectrophotometer. The data are presented as the means ± SD and were evaluated using Student’s *t*-test (*n* = 3). (**D**) Cell cycle analysis of GDF15-knockdown and shCtrl cells by flow cytometry and the percentage of cells in sub-G0/G1, G0/G1, S and G2/M phases are annotated in each column. Data are presented as the means ± SD and were evaluated using Student’s *t*-test (*n* = 3). (**E**) Western blotting analysis showing the expression of cell-cycle-related proteins. NS, not significant. * *p* < 0.05, ** *p* < 0.01, and *** *p* < 0.001.

**Figure 3 ijms-24-02925-f003:**
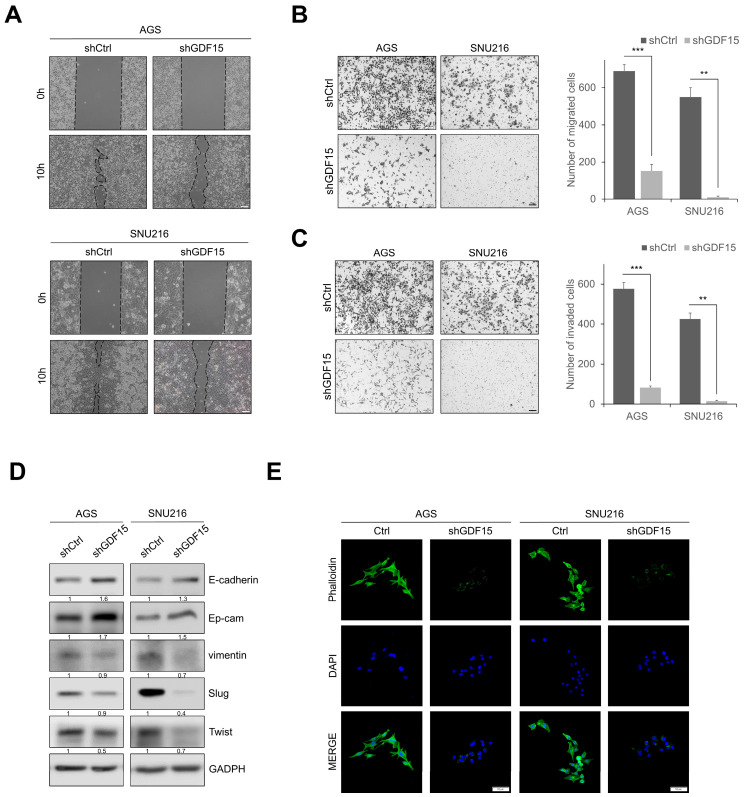
Knockdown of GDF15 suppresses cell migration and invasion. (**A**) Gap closure assay in GDF15-knockdown and shCtrl cells at 0 and 10 h. Scale bars, 100 μm. (**B**,**C**) Transwell migration and invasion assays in GDF15-knockdown and shCtrl cells. Cells were counted in five randomly selected fields. Data are presented as the means ± SD and were evaluated using Student’s *t*-test (*n* = 3). Scale bars, 100 μm. (**D**) Western blotting analysis showed increased expression of epithelial markers and decreased expression of mesenchymal markers in GDF15-knockdown cells compared with shCtrl cells. (**E**) Immunofluorescence images of phalloidin staining (green) of actin stress fibers counterstained with DAPI (blue) in GDF15-knockdown and shCtrl cells. Scale bar, 50 µm. ** *p* < 0.01, and *** *p* < 0.001.

**Figure 4 ijms-24-02925-f004:**
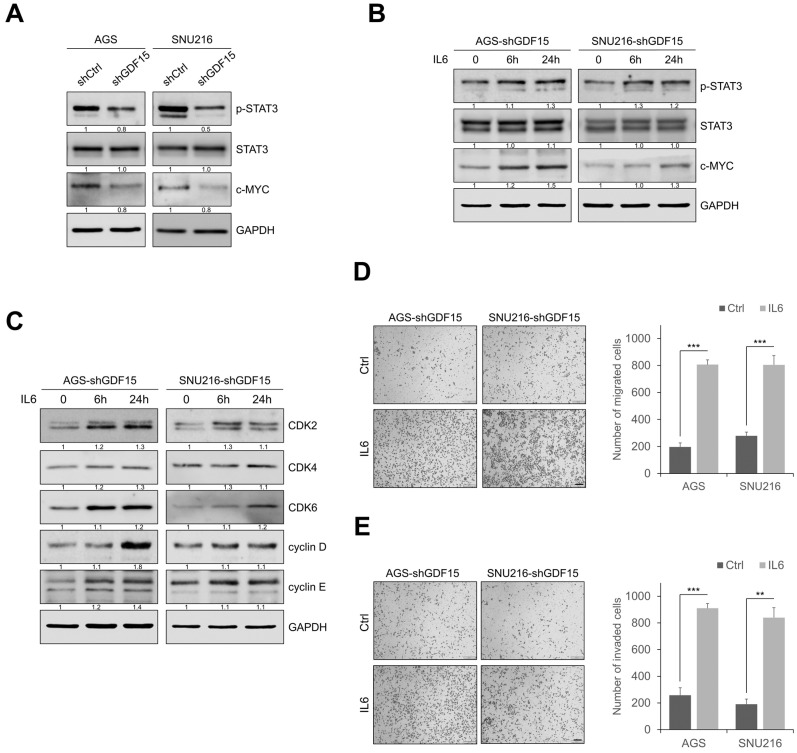
GDF15 knockdown suppresses STAT3 activation. (**A**) Western blotting analysis showing decreased expression of phosphorylated STAT3 and c-MYC in GDF15-knockdown cells compared with shCtrl cells. (**B**) GDF15-knockdown cells were treated with 200 ng/mL of IL-6 for 24 h. Western blotting analysis showed that IL-6 significantly increased the expression of phosphorylated STAT3 and c-MYC. (**C**) Western blot showing the expression of cell-cycle-related proteins in GDF15-knockdown cells treated with IL-6. (**D**,**E**) In transwell assays, migration and invasion were increased in GDF15 knockdown cells treated with IL-6. Cells were counted in five randomly selected fields. Data are presented as the means ± SD and were evaluated using Student’s *t*-test (*n* = 3). Scale bars, 100 μm. ** *p* < 0.01, and *** *p* < 0.001.

**Figure 5 ijms-24-02925-f005:**
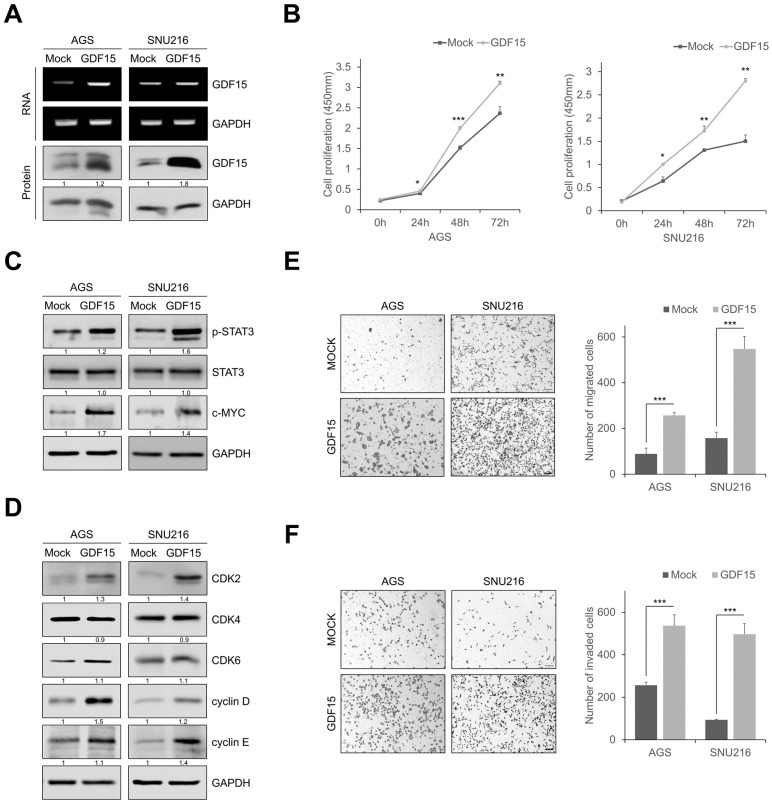
GDF15 overexpression promotes cell proliferation, migration, and invasion by upregulating STAT3 signaling. (**A**) Western blotting analysis and RT-qPCR for GDF15 in GDF15-overexpressing cells. (**B**) In GDF15-overexpressing cells, the cell proliferation rate by the cell counting assay and absorbance was measured at 450 nm using a spectrophotometer. Data are presented as the mean ± SD and were evaluated using Student’s *t*-test (*n* = 3). (**C**) Western blot analysis showing an increased expression of phosphorylated STAT3 and c-MYC in GDF15-overexpressing cells. (**D**) Western blotting analysis showing the expression of cell-cycle-related proteins. (**E**,**F**) Transwell migration and invasion assays of GDF15-overexpressing cells. Cells were counted in five randomly selected fields. Data are presented as the mean ± SD and were evaluated using Student’s *t*-test (*n* = 3). Scale bars, 100 μm. * *p* < 0.05, ** *p* < 0.01, and *** *p* < 0.001.

**Table 1 ijms-24-02925-t001:** Associations of the expression of GDF15 in patients with gastric cancer.

GDF15 Expression					
	Low (*n* = 132)		High (*n* = 46)		
	No.	%	No.	%	*p*
Age, years					
<60 ≥60	66 66	50.0 50.0	19 27	41.3 58.7	0.309
Gender					0.229
Male Female	91 41	68.9 31.1	36 10	78.3 21.7	
Depth of invasion					0.002
T1/2 T3/4	107 25	81.1 18.9	27 19	58.7 41.3	
Nodal involvement					0.003
No Yes	87 45	65.9 34.1	19 27	41.3 58.7	
Stage					0.010
I/II III/IV	110 22	83.3 16.7	30 16	65.2 34.8	
Differentiation					0.089
Differentiated Undifferentiated	68 64	51.5 48.5	17 29	37.0 63.0	
Lymphatic invasion					0.050
No Yes	36 96	27.3 72.7	6 40	13.0 87.0	
Venous invasion					0.120
No Yes	35 97	26.5 73.5	7 39	15.2 84.8	
Tumor size, cm					0.049
<5 ≥5	112 20	84.8 15.2	33 13	71.7 28.3	

**Table 2 ijms-24-02925-t002:** Univariate and multivariate analyses of the clinicopathological parameters and GDF15 expression in patients with gastric cancer.

Variables	Univariate		Multivariate	
	HR (95% CI)	*p*	HR (95% CI)	*p*
Age (≥60 vs. <60 years)	2.097 (1.367–3.218)	0.001	1.886 (1.224–2.905)	0.004
Gender (male vs. female)	0.793 (0.497–1.263)	0.329		
Depth of invasion (T3/4 vs. T1/2)	5.079 (3.316–7.779)	<0.001	1.799 (0.813–3.980)	0.147
Nodal involvement (yes vs. no)	1.792 (1.190–2.699)	0.005	0.554 (0.302–1.018)	0.057
Stage (III/IV vs. I/II)	5.564 (3.600–8.600)	<0.001	4.302 (1.761–10.510)	0.001
Differentiation (differentiated vs. undifferentiated)	1.341 (0.888–2.026)	0.164		
Lymphatic invasion (yes vs. no)	1.290 (0.778–2.139)	0.323		
Venous invasion (yes vs. no)	1.165 (0.710–1.913)	0.546		
Tumor size (≥5 vs. <5 cm)	2.986 (1.905–4.681)	<0.001	1.495 (0.892–2.905)	0.127
GDF15 expression (high vs. low)	1.839 (1.187–2.849)	0.006	1.589 (1.004–2.517)	0.048

## Data Availability

The data presented in this study are contained within the article and the Appendix A.

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
