# Peer review of "GDF15 Promotes Cell Growth, Migration, and Invasion in Gastric Cancer by Inducing STAT3 Activation"

_ijms, 2023, doi:10.3390/ijms24032925_

Round 1

Reviewer 1 Report

It is a very interesting topic and an area of advancing knowledge.

Results are significant but suggest presenting them systematically especially for the first section - comparison between normal and cancer patients, histochemical analysis in gastric cancer patients and then clinical correlates with poor survival.

Also include the baseline characteristics of the CNUH cohort (demographics, tumor characteristics and data on follow up) in a supplementary table.

The histochemical molecular analyses are well presented through stepwise series of logical procedures.

Figure 4 E foot note needs modification/ clarification.

Line 257 “According to our data, GDF15 knockdown inhibited STAT3 activation, but not PI3K/AKT, JNK, or p38 MAPK activation” I am not sure this can be concluded from the presented data. GDF15 knockdown inhibited STAT3 activation but others were not looked at.

Line 263 “In our results, we found that GDF15 mediated inhibition of STAT3 expression reduced the proliferation of gastric cancer cells”. Incorrect statement.

Line 278 “The present study demonstrated that GDF15-mediated STAT3 activation induced EMT in gastric cancer.” Please clarify. EMT markers were not shown in the study.

Line 283 “Taken together, these results suggest that GDF15 induces cell cycle arrest in gastric cancer and inhibits tumor growth, migration, and invasion through STAT3/MYC.” Incorrect statement.

Methodology: Please provide details of the comparative normal patients.

Author Response

It is a very interesting topic and an area of advancing knowledge.

First, thank you for your comments. Your comments were highly insightful and enabled us to greatly improve the quality of our manuscript.

Point 1: Results are significant but suggest presenting them systematically especially for the first section - comparison between normal and cancer patients, histochemical analysis in gastric cancer patients and then clinical correlates with poor survival.

Response 1: In accordance with the reviewer’s suggestion, we have rearranged the first section (Figure 1). Comparison between normal and cancer patients (A-D), histochemical analysis in gastric cancer patients (E) and then clinical correlates with poor survival (F-G).

Point 2: Also include the baseline characteristics of the CNUH cohort (demographics, tumor characteristics and data on follow up) in a supplementary table. The histochemical molecular analyses are well presented through stepwise series of logical procedures.

Response 2: We have added a table of basic characteristics of the CNUH cohort in Supplementary Table 2.

Point 3: Figure 4 E foot note needs modification/ clarification.

Response 3: Based on the reviewer's comments, we have modified the footnote to Figure 4E.

Point 4: Line 257 “According to our data, GDF15 knockdown inhibited STAT3 activation, but not PI3K/AKT, JNK, or p38 MAPK activation” I am not sure this can be concluded from the presented data. GDF15 knockdown inhibited STAT3 activation but others were not looked at.

Response 4: Thanks for your helpful comments. The sentence has been corrected as follows; “According to our data, GDF15 knockdown inhibited STAT3 activation, but not AKT, ERK, p38 and NF- κB activation.”

Point 5: Line 263 “In our results, we found that GDF15 mediated inhibition of STAT3 expression reduced the proliferation of gastric cancer cells”. Incorrect statement.

Response 5: We agree with your comments. The sentence has been corrected as follows; "In our results, we found that GDF15 inhibition induced cell cycle arrest of gastric cancer cells by inhibiting STAT3 activation."

Point 6: Line 278 “The present study demonstrated that GDF15-mediated STAT3 activation induced EMT in gastric cancer.” Please clarify. EMT markers were not shown in the study.

Response 6: Our results confirmed differences in EMT markers in 3D. When GDF15 was reduced, epithelial markers such as E-cadherin and Ep-cam were upregulated in GDF15-knockdown cells, whereas the expression of mesenchymal markers such as vimentin, Slug and Twist were downregulated in shGDF15 cells compared to shCtrl cells. In addition, we found that epithelial markers were downregulated in GDF15-overexpressing cells, whereas expression of mesenchymal markers was upregulated in GDF15-overexpressing cells compared to mock cells (data not shown). And Supplementary information Figure S2E confirmed the expression of EMT markers via siSTAT3 in GDF15 overexpressing cell lines.

Point 7: Line 283 “Taken together, these results suggest that GDF15 induces cell cycle arrest in gastric cancer and inhibits tumor growth, migration, and invasion through STAT3/MYC.” Incorrect statement.

Response 7: The manuscript has been revised based on the comments of the reviewers. The sentence has been corrected as follows; The sentence has been corrected as follows; "Taken together, our study demonstrated that GDF15 affects the progression of gastric cancer, and GDF15 is involved in cell proliferation, migration, and invasion in gastric cancer via STAT3/MYC signaling."

Point 8: Methodology: Please provide details of the comparative normal patients.

Response 8: According to the reviewer's opinion, an experimental method using normal patient tissue was added in western blot analysis and mRNA expression analysis. We have now added the manuscript.

Reviewer 2 Report

The topic of this manuscript lacks the potential interest to the field, and unfortunately there are numerous significant flaws and concerns with this manuscript. Key examples are as follows:

1. There is no in vivo models to support findings.

2. The patient cohort of samples 4 is very small considering the high heterogeneity of gastric cancer, with key information missing (H pylori status, intestinal versus diffuse type etc).

3. All the assays were only performed in 2 cell lines, and the other cell lines were not tested the expression of GDF15.

4. The quality of pictures is poor, particularly in the cell assays.

5. The mechanism of GDF15 regulating STAT lacks the verification.

Author Response

The topic of this manuscript lacks the potential interest to the field, and unfortunately there are numerous significant flaws and concerns with this manuscript. Key examples are as follows:

First, thank you for your comments. Your comments were highly insightful and enabled us to greatly improve the quality of our manuscript.

Point 1: There is no in vivo models to support findings.

Response 1: At the initial stage of cancer research, most studies begin the in vitro model and clinical data analysis. We agree your point that our in vitro data should be validated in the in vivo models. However, the deadline of the revision is very limited (within 10 days). In addition, we believe that our current findings of GDF15 function in tumorigenesis of gastric cancer through STAT3 activation are very timely. Thus, we are asking to the Reviewer to understand our situation and to exempt the in vivo experiments in the present study. Your valuable comments will be tested in our future studies and we hope you understand our meaning.

Point 2: The patient cohort of samples 4 is very small considering the high heterogeneity of gastric cancer, with key information missing (H pylori status, intestinal versus diffuse type etc).

Response 2: Following the reviewer's suggestion, we added the information of patient to the Supplementary Table 1.

Point 3: All the assays were only performed in 2 cell lines, and the other cell lines were not tested the expression of GDF15.

Response 3: Thanks for your helpful comments. Initially, we planned to use both gain-of-function and loss-of-function approaches. And the expression of GDF15 was confirmed in 11 gastric cancer cell lines (MKN1, MKN28, MKN74, SNU1, SNU16, SNU216, SNU638, SNU668, AGS, NCI-N87, Hs-746T) using western blot analysis (as shown in the figure below, data not shown). Most of the cells expressed GDF15, especially SNU216, SNU638, and AGS cells.

However, when stable cell lines were established and confirmed by western blot analysis, AGS and SNU216 cells showed more distinct differences (as shown in the figure below, data not shown). Therefore, we conducted experiments using 2 cell lines (ASG, SNU216).

Point 4: The quality of pictures is poor, particularly in the cell assays.

Response 4: At the suggestion of reviewer, we changed Figure 3E with better cell assays pictures.

Point 5: The mechanism of GDF15 regulating STAT lacks the verification.

Response 5: Thank you for your instructive comments. In this study, it was confirmed that the expression of STAT3 was down-regulated by GDF15 to suppress aggressive biological behavior. In addition, previous studies have confirmed that GDF15 regulates STAT3 in thyroid cancer and Glioma stem cells (Kang, Y.E.; et al. Growth Differentiation Factor 15 is a Cancer Cell-Induced Mitokine That Primes Thyroid Cancer Cells for Invasiveness. Thyroid 2021, 31, 772-786, doi:10.1089/thy.2020.0034 and Zhu, S.; et al. GDF15 promotes glioma stem cell-like phenotype via regulation of ERK1/2-c-Fos-LIF signaling. Cell Death Discov 2021, 7, 3, doi:10.1038/s41420-020-00395-8.). However, the exact mechanism by which GDF15 downregulates STAT3 expression in gastric cancer remains unclear. Therefore, we plan to conduct further studies to elucidate the molecular mechanism of GDF15 regulation by STAT3 and hope you understand our meaning.

Round 2

Reviewer 2 Report

1. The animal mode in vivo and the mechanism of GDF15/STAT3 regulation is very important.

2. Please explain the reason of the STAT3 as the target of the GDF15. Once you have further study the STAT3, why do you screened several signaling molecules.

3. Supply the all original WB figures.

4. Pay more attention to the writing format of the manuscript, for example, line 58. The reference should in the end of the sentence.

5. Polish the language by a professional editor.

Author Response

Point 1: The animal mode in vivo and the mechanism of GDF15/STAT3 regulation is very important.

Response 1: We agree with you. As we stated in the previous responses, the duration of revision is very short, not enough to perform in vivo experiments. Thus, we have included the comments about the limitation of the study in the Discussion section (lines 290-295), as follows:

Although our data has a limitation of the lack of in vivo models, clinical data and in vitro studies strongly suggest that GDF15 promotes the progression of gastric cancer. In addition, our study demonstrated that GDF15 is involved in cell proliferation, migration, and invasion in gastric cancer via STAT3/MYC signaling. Future studies are needed to characterize in vivo function of GDF15 in tumorigenesis of gastric cancer.

Point 2: Please explain the reason of the STAT3 as the target of the GDF15. Once you have further study the STAT3, why do you screened several signaling molecules.

Response 2: The reason why we focused on GDF15-STAT3 axis is based on the previous findings (#31, and 32), which reported the GDF15-mediated STAT3 signaling in other tumor models including thyroid cancer and glioma stem cells in the manuscript. We thus asked whether the same signaling axis is functional in gastric cancer cells. We have included the description (lines 66-68), as follows:

Given the findings that GDF15 is required for the activation of STAT3 and tumorigenesis in thyroid cancer and glioma stem cells [31,32], we focused whether GDF15 function is mediated through STAT3 signaling in gastric cancer cells.

And at the beginning of this study, we screened several signaling molecules in different ways. Thus, we have included the comment about why we screened several signaling molecules in the Result section (lines 170-180), as follows:

We next investigated the mechanisms by which GDF15 induces carcinogenesis in gastric cancer cells. Previous studies reported that GDF15 is required for the activation of STAT3 and tumorigenesis in thyroid cancer and glioma stem cells [31,32]. We thus examined whether the blockade of GDF15 affected the STAT3 activation in gastric cancer cells. We observed that GDF15 knockdown decreased the phosphorylation of STAT3 and c-MYC (Fig. 4A). In addition, there are several signaling pathways including AKT, ERK, p38 and NF-κB, are regulated by GDF15 in other cancers [33,34]. When we further screened these signaling pathways are perturbed in GDF15-knockdowned gastric cancer cells, we found that the activation of AKT, ERK, p38, and NF-κB was not different between shCtrl and GDF15-knockdown gastric cancer cells (Fig. S2A). These data suggest that GDF15-mediated tumorigenesis is mainly mediated through STAT3, but not these signaling pathways in gastric cancer cells.

Point 3: Supply the all original WB figures.

Response 3: According to the reviewer's suggestion, I would like to attach the original WB figures again, but I cannot attach them as an Excel file. Previously, we have submitted original WB figures along with all source data and all original figures. The editorial office said they would send it directly to the reviewer.

Point 4: Pay more attention to the writing format of the manuscript, for example, line 58. The reference should in the end of the sentence.

Response 4: The manuscript was revised based on the reviewer’s comments.

Point 5: Polish the language by a professional editor.

Response 5: Following the reviewer's suggestion, we had the language proofread by a professional editor. (Please refer to the attached response letter.)

Round 3

Reviewer 2 Report

Accepted.